# Metal Oxide Nanorods-Based Sensor Array for Selective Detection of Biomarker Gases

**DOI:** 10.3390/s21051922

**Published:** 2021-03-09

**Authors:** Gwang Su Kim, Yumin Park, Joonchul Shin, Young Geun Song, Chong-Yun Kang

**Affiliations:** 1KU-KIST Graduate School of Converging Science and Technology, Korea University, Seoul 02841, Korea; 218307@kist.re.kr (G.S.K.); wsx6659@kist.re.kr (Y.P.); 2Electronic Materials Research Center, Korea Institute of Science and Technology (KIST), Seoul 02791, Korea; spy21318@kist.re.kr

**Keywords:** semiconducting gas sensor, array, nanostructure, metal oxide

## Abstract

The breath gas analysis through gas phase chemical analysis draws attention in terms of non-invasive and real time monitoring. The array-type sensors are one of the diagnostic methods with high sensitivity and selectivity towards the target gases. Herein, we presented a 2 × 4 sensor array with a micro-heater and ceramic chip. The device is designed in a small size for portability, including the internal eight-channel sensor array. In_2_O_3_ NRs and WO_3_ NRs manufactured through the E-beam evaporator’s glancing angle method were used as sensing materials. Pt, Pd, and Au metal catalysts were decorated for each channel to enhance functionality. The sensor array was measured for the exhaled gas biomarkers CH_3_COCH_3_, NO_2_, and H_2_S to confirm the respiratory diagnostic performance. Through this operation, the theoretical detection limit was calculated as 1.48 ppb for CH_3_COCH_3_, 1.9 ppt for NO_2_, and 2.47 ppb for H_2_S. This excellent detection performance indicates that our sensor array detected the CH_3_COCH_3_, NO_2_, and H_2_S as biomarkers, applying to the breath gas analysis. Our results showed the high potential of the gas sensor array as a non-invasive diagnostic tool that enables real-time monitoring.

## 1. Introduction

Healthcare is now undergoing a paradigm shift that will transform the nature of medicine from reactive to preventive [1]. The changes are facilitated by a new approach to a disease that will lead to the emergence of personalized medicine that focuses on the nearby environment, treatment, and disease prevention in individual patients [2]. Large-scale healthcare methods, which have relied on waiting for the patient to get sick, will be replaced by personalized, predictive, preventive, and participatory (P4) medicine via the convergence of the approaches to disease, useful measurement, visualization techniques, and new computational tools [3].

Based on these needs, monitoring the harmful gases present in the environment or breathing has been extensively concentrated as a suitable healthcare approach [4]. In general, indoor air quality is contaminated by harmful gases emitted from building materials, furniture, and appliances, which induce dizziness, paralysis, dyspnea and eventually lead to a comatose state [5]. These gases contain a lot of chemical vapors (NO_2_, CO, NH_3_, and volatile organic compounds (VOCs)) at different concentrations ranging from parts per billion (ppb) to parts per million (ppm) [6]. Furthermore, some chemical vapors, by-products of metabolic processes, show apparent correlation with a particular disease and possibly indicate potential diseases such as asthma [7], diabetes [8], liver diseases [9], lung cancer [10], and metabolic disorders [11], working as biomarkers. However, commercial instruments such as gas chromatography-mass spectrometry (GC-MS) and selected ion-flow tube mass spectrometry (SIFT-MS) are too bulky and costly to use anytime and anywhere [12]. They could also not offer continuous or real-time information regarding the present state of indoor or individual patients [13].

To overcome these drawbacks, many researchers have studied metal oxide semiconductors (MOSs) due to their outstanding advantages, such as low cost, simplicity in fabrication, high sensitivity, easy integration with electronic circuits, and a large number of detectable gases [14]. Cutting-edge research has been spotlighted to apply the nanostructured materials [15], catalysts [16], heterojunctions [17], and UV activation [18] to the sensors for the enhancement of gas sensing performance. However, relatively low gas selectivity remains a challenge to be solved [19]. In general, the catalysts, including Au [20], Pt [21], and Pd [22], functionalize the surface of the sensing materials, leading to an enhancement of sensitivity and selectivity via electronic and chemical sensitization, respectively. However, there are limits to classifying various gases with a combination of a single catalyst and metal oxide. For directly comparing and analyzing the role of the catalyst and classifying multiple gases, the sensor array is considered a suitable approach [23].

Herein, we presented a 2 × 4 sensor array using In_2_O_3_ and WO_3_ nanorods (NRs), decorated with Au, Pt, and Pd utilizing an electron beam evaporator based on our previous reports [24] (In_2_O_3_ NRs, Au-decorated In_2_O_3_ NRs, Pt decorated In_2_O_3_ NRs, Pd decorated In_2_O_3_ NRs, WO_3_ NRs, Au-decorated WO_3_ NRs, Pt decorated WO_3_ NRs, and Pd decorated WO_3_ NRs). We then evaluated the 2 × 4 array gas sensor with the maximum detection performance through a metal catalyst for detection characteristics at an operating temperature of 150–300 °C toward CH_3_COCH_3_, NO_2_, and H_2_S, which are representative biomarkers of diabetes, asthma, and halitosis, respectively.

## 2. Materials and Methods

### 2.1. Device Fabrication

The sensor array composed of Pt/Ti (80 nm/20 nm thick) interdigitated electrodes (IDEs) was fabricated on a four-inch SiO_2_/Si wafer through standard photolithography and lift-off processes. The distances between each electrode were 5 μm, and there were 20 electrodes in a 1 mm × 0.25 mm area. An electron-beam evaporator was subsequently used to deposit 300 nm-thick In_2_O_3_ and WO_3_ nanostructures on the prepared IDEs at a glancing angle of 80° in off-axis mode. The substrate was placed 30 cm away from the crucible, and the lift-off process was employed to deposit only onto the area containing the electrodes. As a next step, the catalysts, including Pt, Au, and Pd, were evaporated on the metal oxides by electron beam evaporator in on-axis mode. All the fabricated sensor arrays were annealed at 550 °C for 2 h in ambient air to crystallize the metal oxides and agglomerate the catalysts. The annealing temperature was applied considering the trade-off relationship between the sensor stability and gas response. Then, the sensor array consisting of the 2 × 4 sensor array was mounted on an Ag-based micro-heater and a chip carrier using a silver paste and ceramic bond, respectively.

### 2.2. Characterization

X-ray diffraction (DMax2500) was used to analyze deposited films with 2θ scan from 20° to 50°, where Cu Kα radiation (wavelength; 1.5418 Å) was used for the X-ray source with a fixed incident angle of 2°. The morphology of the fabricated nanostructures was observed using a field-emission scanning electron microscope (SEM) (Inspect F50) with an acceleration voltage of 15 kV and a working distance of 10 mm.

### 2.3. Sensor Property Measurement

We measured the gas sensing properties of the 2 × 4 sensor array in a hand-made measuring chamber. The operating temperature was controlled by heating the micro-heater using a programmable power supply (PSH-3620A) and was calibrated using an infrared camera. The gas flow rate of 1000 sccm was maintained using mass-flow controllers. In this condition, the gas was changed from dry air to the calibrated target gas (balanced with dry air). The resistance was measured at a dc bias voltage of 1 V using a source measurement unit (Keithley 2401), and each sensor was connected using a switch system (Keithley 7001). All the measurements were recorded on a computer using LabVIEW with general purpose interface bus (GPIB).

## 3. Results and Discussion

Highly porous nanostructured metal oxides have been mainly used as gas sensing materials because of their large surface-to-volume ratio, narrow necks between each grain, and effective gas accessibility. In our previous work [25], the glancing angle deposition (GAD) method was reported to fabricate useful gas sensing materials. As shown in Figure 1a, the 300-nm thick nanorods metal oxide including (In_2_O_3_ and WO_3_) was evaporated on the Pt interdigitated electrodes (IDEs) using the GAD method at a glancing angle of 80°. When depositing the metal oxides, the incident angle of the vapor flux generates the self-shadowing effect, resulting in the highly porous nanostructured materials (Figure 1b). Subsequently, the catalysts, including Au, Pt, and Pd was evaporated on the surface of the metal oxides with different thickness of 1–2 nm (Figure 1c). The Au, Pt, and Pd were deposited under the condition of becoming a uniform 1 or 2 nm-thick film. To reduce the deposition and photolithography sequences, we chose the 2 × 4 array, which maximizes the sort of gas sensor up to eight sensors. The 2 × 4 array sensor can drastically reduce the number of deposition and photolithography processes required for sensor fabrication. The identical sensing materials and catalysts are deposited for the vertical and horizontal directions, respectively. All fabricated sensor array was annealing at 550 °C for 2 h to crystallize the metal oxides. The catalysts were aggregated to nanoparticles, resulting in the metal oxide NRs decorated with the catalyst nanoparticles on the surface. Subsequently, the 2 × 4 sensor array was mounted on the Au-based micro-heater for back heating and the chip carrier for electrical connecting (Figure 1d).

To investigate the morphologies of the sensing materials, the SEM was carried out as shown in Figure 2a–d,f–i. Insets in Figure 2 indicate cross sectional SEM images. The highly porous and well aligned In_2_O_3_ NRs and WO_3_ NRs were observed in Figure 2a,f. When decorating the catalysts on the metal oxide surface, the nano size of nanoparticles was observed in Figure 2 b–d,g–i. XRD in Figure 2e,j characterized the crystallinities of the 2 × 4 sensors. There is no significant impurity phase from the XRD results, indicating that the In_2_O_3_ NRs and WO_3_ NRs were well crystallized during the annealing process. At the same time, peaks corresponding to Au, Pt, and Pd were not observed. We assume that the peaks corresponding to Au, Pt, and Pd are underneath those corresponding to In_2_O_3_ and WO_3_ because the surface contains less catalyst components. We presented SEM images of a metal catalyst on a SiO_2_ substrate. As shown Appendix A, the metal catalysts of Au, Pt, and Pd were clearly deposited on the surface and uniformly dispersed in the form of nanoparticles on the SiO_2_ substrate. Compared with Figure 2 in the manuscript, the metal nanoparticles deposited on the SiO_2_ thin film shows relatively large diameter-sizes because of a lower surface energy of the SiO_2_. Therefore, we expected that the metal catalysts were well dispersed on the surface of oxide nanorods with nanosized particles.

The gas sensing properties of the 2 × 4 sensor array were measured in the sensing chamber shown in the Figure 3a, and the array signals were acquired simultaneously in real time. We fabricated a measuring chamber for an array. The chamber was sealed with rubber to prevent the inflow of external atmosphere and leakage of the internal measurement atmosphere. The metal oxide-based semiconductor gas sensor has been focused on because of its inexpensiveness, small size, and ease of integration with electronic circuits. However, it requires a high operating temperature of about 200–400 °C to maximize the gas response and accessibility, resulting in high power consumption and poor sensor stability because of thermally induced grain growth. Recently, there are many reports to reduce the operating temperature and the power consumption by miniaturizing the gas sensor. In this regard, we can significantly reduce the power consumption of our array sensor by miniaturizing the chip size. The operating temperature was controlled by back heating the micro-heater. The power consumption of the micro-heater is indicated in Figure 4a. By increasing the power consumption of the micro-heater, the operating temperature linearly increased. Although our micro-heater requires massive power consumption compared to that of the conventional gas sensors, an energy consumption of the heater is sufficiently reduced by miniaturizing the sensor array and decreasing the heating region.

The sensing properties of the semiconducting gas sensor are highly affected by the operating temperature; with an increase in the temperature, the gas adsorption rate increases due to reduced activation energy with the sensing materials, resulting in the enhancement of the gas response. However, the gas response tends to decrease above a certain temperature since the desorption rate also increases. Therefore, the operating temperature optimization according to the target gases is essential for the semiconducting gas sensors. We selected CH_3_COCH_3_, NO_2_, and H_2_S as representative biomarkers of diabetes, asthma, and halitosis, respectively [26]. Figure 5a–c show the 2 × 4 sensors’ response to CH_3_COCH_3_, NO_2_, and H_2_S gas over a temperatures range of 150–350 °C with a setting span of 50 °C for the precise temperature classification and efficient experimentation. The slight modulation of gas flow rate can influence on the operating temperature although a constant flow rate was set to 1000 sccm in this experiment. However, the base resistances of all sensors are not significantly changed for the measurement, indicating that the operating temperature is identical or is less affected by the slight modulation of flow rate. The response of the sensor is defined as R_a_/R_g_ − 1 in reducing gas and R_g_/R_a_ − 1 in oxidizing gas, depending on the gas type [27]. R_a_ indicates the sensor resistance in the absence of gas, and R_g_ denotes the resistance in the presence of target gas. In the case of CH_3_COCH_3_, the overall detection performance was excellent at 300 °C, and Au-decorated In_2_O_3_ NRs and Pt-decorated In_2_O_3_ showed the highest response compared to other sensors (Figure 5a). On the other hand, NO_2_ shows the highest detection performance at 150 °C, and Au-decorated In_2_O_3_ NRs show the highest response (Figure 5b). For H_2_S, Au-decorated WO_3_ NRs showed high detection characteristics at 250 °C (Figure 5c). The optimized temperature for the gas sensors is determined by the adsorption as mentioned above and desorption rates depending on the target gas. Moreover, the oxygen species on the surface of the metal oxide can affect the optimum temperature. With increasing the temperature, the negatively charged oxygen species are generated. It is known that the ionosorption (O_2_^−^) of physisorbed oxygen is formed at more than 100 °C. Subsequently, the oxygen is chemisorbed on the surface as a form of O^−^ up to 350 °C. Above the temperature of 350 °C, O_2_^−^ is generated by extracting the electrons from the metal oxides. Based on these reactions, NO_2_ competes with O_2_ to adsorb on the metal oxide surface, indicating a lower response at a high operating temperature [28]. In contrast, the reducing gases involving CH_3_COCH_3_ and H_2_S show a higher response at high temperature. Therefore, we adopted optimum temperatures of 300, 150, and 250 °C towards CH_3_COCH_3_, NO_2_, and H_2_S, respectively. The high response of our sensor array was elucidated by utility factor, transducer function, and receptor function. The narrow necks between each NRs can enhance the response to target gas (Figure 5d), and the high porosity enables effective target gas accessibility (Figure 5e) [29]. The use of catalysts in the gas sensors has been considered as a promotor of gas adsorption by a spill-over effect (Figure 5f) [30].

To examine the response linearity and detection limit of our sensor, we exposed the 2 × 4 sensor array to 100–500 ppb CH_3_COCH_3_, NO_2_, and H_2_S at the operating temperatures of 300, 150, and 250 °C, respectively. As mentioned above, the gas sensing properties were evaluated at the optimum temperatures depending on the target gas. The Figure 6a–c shows response transients of eight sensors. Based on these results, the responses are plotted in Figure 6d–f. Upon exposure to target gases, the responses linearly increase with increasing the gas concentration. Although the 100 ppb was the lowest concentration examined experiment in the study, the theoretical detection limits (signal-to-noise ratio > 3) are calculated to be 1.48 ppb, 1.90 ppt, and 2.47 ppb toward CH_3_COCH_3_, NO_2_, and H_2_S, respectively. The following result refers to the potential of the gas sensor array, which can trace the meager difference of the gas concentration.

In a semiconducting gas sensor, it is difficult to clearly realize gas selectivity because the characteristics of the sensor vary depending on the combination of catalyst and metal oxide, the morphology of the nanostructure, and the deposition method.

To clearly demonstrate the selectivity of our sensor array, we plotted the polar plots. Figure 7 shows the response of 2 × 4 array sensor to CH_3_COCH_3_ 10 ppm, NO_2_ 1 ppm, and H_2_S 1 ppm at the optimized temperature (300, 150, and 250 °C). For the Figure 7a,b, the single target detection of CH_3_COCH_3_ and NO_2_ was clearly observed because both gases are highly dependent on the temperature (Figure 5a,b). Furthermore, the selective detection at 250 °C in Figure 7c is achieved by different patterns with each target gas.

To develop our sensor array for medical applications, it is necessary to investigate the effect of water molecules on the gas sensing properties because of high relative humidity in exhaled gas [31]. In addition, it is reported that the exhaled gas contains a myriad of volatile organic compounds (VOCs), which hinders single target detection of a biomarker [32]. Therefore, we plan to improve the gas selectivity using deep learning-based data analysis and precise analysis, including primary component analysis (PCA) [33], factor analysis (FA) [34], and cluster analysis (CA) [35].

## 4. Conclusions

In the recent healthcare market, a portable diagnostic device can reduce the inconvenience of visiting a hospital for disease prevention and diagnosis. [36] The electronic nose, which diagnoses diseases by detecting biomarkers in exhaled gas, is being studied as a promising candidate. [37] In this study, we developed an array gas sensor device based on nanostructured metal oxide sensing materials. The device consists of a 2 × 4 array gas sensor with eight channels, a micro-heater, and a ceramic chip. We measured the fabricated 2 × 4 array gas sensor for CH_3_COCH_3_, NO_2_, and H_2_S gases, known as biomarkers for human disease. As a result of the measurements, the 2 × 4 array gas sensor showed high detection performance at sub ppb level for target gases and showed excellent recovery and repeatability. Moreover, each channel, separated by a metal catalyst, presented different detection performance according to the measurement gas and was also distinguishable at a glance through a polar plot. The device exhibited high detection performance for CH_3_COCH_3_, NO_2_, and H_2_S gases, with high reproducibility, manufacturing simplicity, and high yield mass production, presenting great potential for various practical applications.

## Figures and Tables

**Figure 1 sensors-21-01922-f001:**
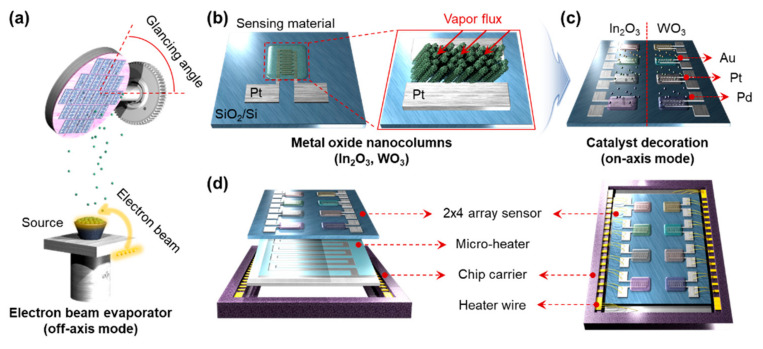
Schematic illustration of (**a**) the fabrication procedures for porous nanostructures using GAD. (**b**) A design of Pt-IDEs and metal oxide nanorods grown the direction of the vapor flux. (**c**) The position of Au, Pt, and Pd catalysts decorated by e-beam evaporator using on-axis mode. (**d**) 2 × 4 sensor array with 2 × 4 sensor array, back heater, chip carrier, and Au wires.

**Figure 2 sensors-21-01922-f002:**
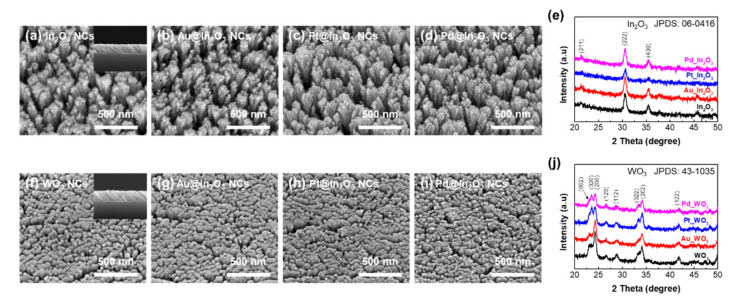
Top-view and cross-sectional (inset) FE-SEM images of (**a**–**d**) bare, Au-, Pt-, and Pd-decorated In_2_O_3_ nanorods, (**f**–**i**) bare, Au- (2 nm), Pt- (1 nm), and Pd- (2 nm) decorated WO_3_ nanorods. X-ray diffraction pattern of (**e**) In_2_O_3_ and (**j**) WO_3_ nanorods as a function of decorated catalysts.

**Figure 3 sensors-21-01922-f003:**
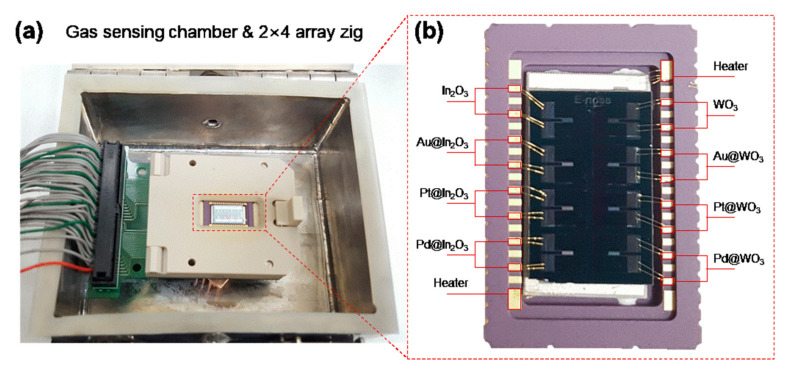
(**a**) Optical image of the gas sensing chamber and (**b**) 2 × 4 sensor array mounted on the micro-heater and the chip carrier.

**Figure 4 sensors-21-01922-f004:**
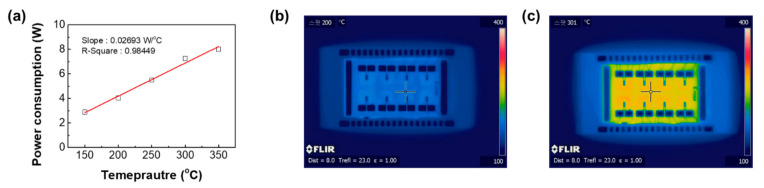
(**a**) Power consumption of the micro-heater. Infrared images of 2 × 4 sensor array with different operating temperature; (**b**) 200 °C and (**c**) 300 °C.

**Figure 5 sensors-21-01922-f005:**
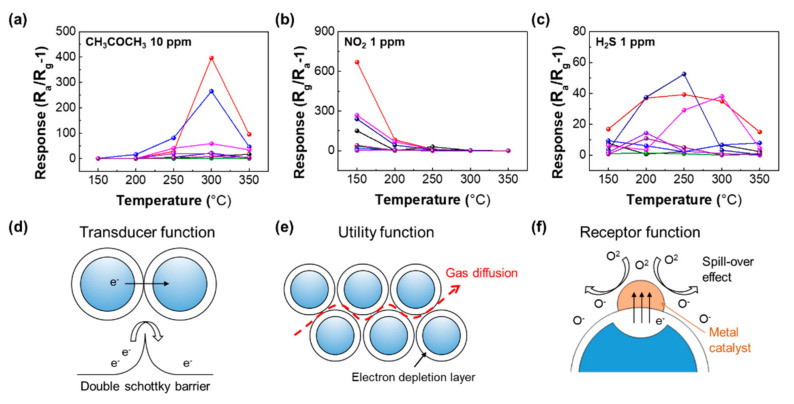
Response of the 2 × 4 sensor array to (**a**) 10 ppm CH_3_COCH_3_, (**b**) 1 ppm NO_2_, and (**c**) H_2_S 1 ppm vs. the wide range of operating temperatures from 150 to 350 °C. (**d**) Transducer function, (**e**) utility function, and (**f**) receptor function and spill-over effect that represent gas mechanisms.

**Figure 6 sensors-21-01922-f006:**
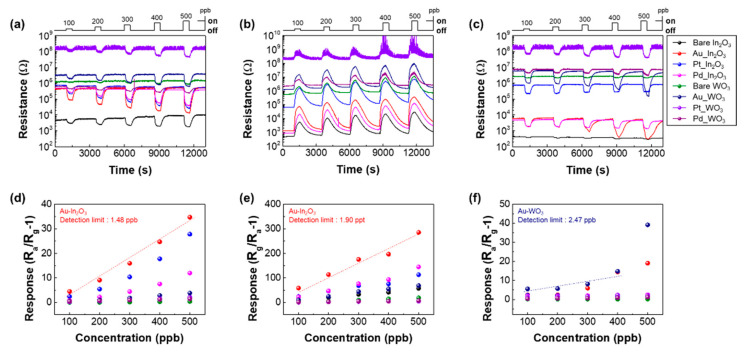
Response of the 2 × 4 sensor array to 100–500 ppb (**a**) CH_3_COCH_3_, (**b**) NO_2_, and (**c**) H_2_S at 300 °C, 150 °C, and 250 °C, respectively. Theoretical detection of limit of 2 × 4 sensor array to 100–500 ppb (**d**) CH_3_COCH_3_, (**e**) NO_2_, and (**f**) H_2_S.

**Figure 7 sensors-21-01922-f007:**
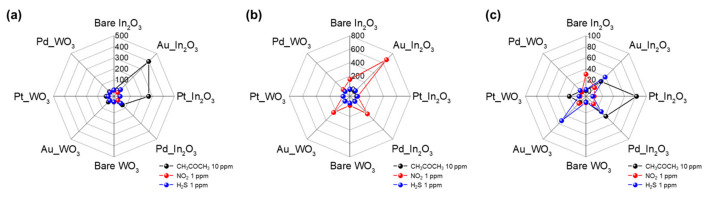
Polar plot of 2 × 4 array sensor responses of (**a**) 10 ppm CH_3_COCH_3_, (**b**) 1 ppm NO_2_, and (**c**) 1 ppm H_2_S at the operating temperatures of 300, 150, and 250 °C.

## Data Availability

The data presented in this study are available on request from the corresponding author. The data are not publicly available to protect the privacy of the subjects.

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
