# Peer review of "Metal Oxide Nanorods-Based Sensor Array for Selective Detection of Biomarker Gases"

_sensors, 2021, doi:10.3390/s21051922_

Round 1
Reviewer 1 Report
1 Why is the temperature setting span of 50 ℃ when the author searches for the best working temperature according to the target gas? What is the reason? 2 Compared with single sensor, how much better is the detection performance of sensor array? It is suggested to show the experiment of single sensor detection. 3 Why did the author choose 2 * 4 sensor array instead of other specifications?Author Response
Dear Reviewer,
We have revised our manuscript entitled “Metal-oxide nanorods based sensor array for selective detection of biomarker gases”, which had been submitted for publication in Sensors as a full paper.
We have elaborately revised the manuscript according to the reviewer’s comments. We attach the response to the reviewer’s comments, and the revised manuscript with the changes highlighted in yellow. Please find the change in the revised manuscript with highlighted in yellow.
Thank you for spending your time and I am looking forward to hearing from you.
Thank you very much.
Sincerely yours,
Chong-Yun Kang

Reviewer 2 Report
The authors report about the design, fabrication and test on an eight-element metal oxide gas sensor array intended for the medical application of breath analysis. Target gases are CH3COCH3 (acetone), NO2, and H2S which are considered as biomarkers for diabetes, asthma and halitosis. The authors have successfully built a 2x4 metal oxide gas sensor array that is able to detect these target gases in the required low concentration ranges.
Suggestions for improvement are the following:
- considering the application of breath analyis, it is obvious that humidity will be a major background gas. In its revised form the paper should provide data on target gas detection with and without background humidity.
- In the introduction section the authors mention background gases which are likely to be present in variable concentrations and mixtures in indoor environments. It is not clear how these gases interfere with the detection of the target gases. An approach towards overcoming such problems is first operating the gas sensor arrays in purified reference air and then exposing it to the breath samples. For the convenience of the authors, I have attached a paper where this method has been successfully employed.
- Breathing across a heated sensor array will alter its operation temperature and thus change the array´s sensitivity towards the target gases --> arrays either need to be operated under no-flow conditions or the array temperature needs to be kept constant by electronic feedback control (paper attached)
- A final question: in case all three target gases will be present, will the output pattern be a linear superposition of those patterns shown in Fig.7 or will it be completely different? Please check!
A minor omprovement concerns Figure 4. This Figure shows five IR images, acquired at five different senor operation temperatures, which very surprisingly all look more or less alike. Because of the limited space, instrumental parameters on these images are hardly readable which makes it difficult to interprete these images. It is suggested to show only one or two of those images, color-coded in a way that highlights differences in the different temperature ranges.
Overall the paper stands well in the competition with similar papers on metal oxide gas sensor arrays. However, common to most companion papers, the present paper falls short in critically assessing cross-sensitivity issues and in describing technical approaches towards overcoming these known shortcomings of MOX gas sensor arrays.
I therefore propose critical experimental checks before the paper is published.

Author Response
Dear Reviewer,
We have revised our manuscript entitled “Metal-oxide nanorods based sensor array for selective detection of biomarker gases”, which had been submitted for publication in Sensors as a full paper.
We have elaborately revised the manuscript according to the reviewer’s comments. We attach the response to the reviewer’s comments, and the revised manuscript with the changes highlighted in yellow. Please find the change in the revised manuscript with highlighted in yellow.
Thank you for spending your time and I am looking forward to hearing from you.
Thank you very much.
Sincerely yours,
Chong-Yun Kang

Reviewer 3 Report
Comments to the author:
In this manuscript, the authors reported the fabrication of a 8-channel sensor array based on In2O3 NRs and WO3 NRs decorated with Pt, Pd, or Au metal catalysts to enhance functionality. The authors evaluated this device by measuring the exhaled gas biomarkers CH3COCH3, NO2, and H2S in order to confirm the respiratory diagnostic performance. Overall, I would not recommend the publication of this contribution as it is because there are some issues to be clarified.
The following are some questions and suggestions for improving their work:
Major issues:
- The key point is that the authors claim to use a sensor array to improve the selectivity of their sensor but at the end they are not taking advantage of this array. The authors just evaluated each gas individually for each sensor. However the target of a array sensor is to dinstinguish a mixture o gases thanks to statistical analysis of the response of the overall array. If the authors are using an array sensor, they need to evaluate the response towards the mixture of this gases. If not I don’t clearly see the goal of using an array.
- From SEM the authors can not demonstrate the evidence that they have the metals (Au, Pt or Pd) on their devices. Perhaps the authors should do a control test without the metal oxides to clearly see the metals (SEM or XRD), otherwise there is no clear evidence along the manuscript of the presence and characterization of this metals.
- The authors did not explain clearly what Ra/Rg-1 refers to and why this is the best manner to evaluate the performance of their sensors.
Minor issues:
- The authors claimed that the metals (Au, Pt or Pd) are minimal size. This is not very accurate. A proper size should be calculated.
- The authors claimed ‘commercial instruments such as gas chromatography-mass spectrometry (GC-MS) and selected ion-flow tube mass spectrometry (SIFT-MS) are too bulky and costly to use any-time and anywhere’. However their sensor operates at temperatures around 300ºC. Is this characteristic a drawback for the commercial use of their sensors?
Author Response

(The authors gave the same response as above.)

Round 2
Reviewer 2 Report
The authors have developed a metal oxide based gas sensor array for the detection of biomarkers for diabetes, asthma and halitosis. The authors have shown that all three gases can be detected and distinguished from each other at the required low concentration levels when provided in a background of dry synthetic air. This achievement is a necessary but not a sufficient condition that the array would actually work in the envisaged application of medical breath analysis.
Accepting that the authors are limited in their laboratory work by the current COVID crisis, I strongly suggest that the authors add a short paragraph at the end of the paper that contains an outline of additional research and development work that needs to be done to arrive at a medical breath analyzer that would be useful in a clinical environment.
Author Response
Dear reviewer,
We have revised our manuscript entitled “Metal-oxide nanorods based sensor array for selective detection of biomarker gases”, which had been submitted for publication in Sensors as a full paper.
We have elaborately revised the manuscript according to the reviewer’s comments. We attach the response to the reviewer’s comments, and the revised manuscript with the changes highlighted in yellow. Please find the change in the revised manuscript with highlighted in yellow.”
Thank you for spending your time and I am looking forward to hearing from you.
Thank you very much.
Sincerely yours,
Chong-Yun Kang

Reviewer 3 Report
Unfortunately the authors did not reply all my concerns.
- The explanation about why the authors are not using the whole sensing capabilities of an array is not convincing. Their sensor is not selective, the authors can not claim that if the analytes can not be distinguished. The authors evaluated different receptor molecules against different analytes but there is no point on making an array, it does not provide any value.
- Regarding the SEM, in the Figure S1 the authors provided it is clear that the Pd size is not 2 nm and neither Au or Pt are 2 and 1 nm respectively. Also the authors are claiming that they expect that the metals are on the oxide nanorods but they can not prove it with their characterization.
- The authors did not properly define Ra/Rg 1 in the text. Although it’s the response equation each of the parameters should be defined.
Author Response

(The authors gave the same response as above.)
